# Cemented Dual Mobility Cup for Primary Total Hip Arthroplasty in Elder Patients with High-Risk Instability

**DOI:** 10.3390/geriatrics6010023

**Published:** 2021-03-07

**Authors:** José María Lamo-Espinosa, Jorge Gómez-Álvarez, Javier Gatica, Álvaro Suárez, Victoria Moreno, Pablo Díaz de Rada, Andrés Valentí-Azcárate, Matías Alfonso-Olmos, Mikel San-Julián, Juan Ramón Valentí-Nin

**Affiliations:** 1Department of Orthopaedic Surgery and Traumatology, Clínica Universidad de Navarra, Pamplona, 31008 Navarra, Spain; jgomeza@unav.es (J.G.-Á.); jgatica85@gmail.com (J.G.); asuarezlope@unav.es (Á.S.); vmfigaredo@unav.es (V.M.); avalenti@unav.es (A.V.-A.); malfonsool@unav.es (M.A.-O.); msjulian@unav.es (M.S.-J.); jrvalenti@unav.es (J.R.V.-N.); 2Department of Orthopaedic Surgery and Traumatology, Hospital Reina Sofía, Tudela, 31005 Navarra, Spain; pderada.pddr@gmail.com

**Keywords:** total hip arthroplasty, cemented dual mobility cup, complications and survival

## Abstract

Several studies have shown that double mobility (DM) cups reduce postoperative dislocations. Does the cemented dual mobility cup reduce dislocations in a specific cohort of elder patients with a high dislocation risk? Our hypothesis is that this implant is optimal for elder patients because it reduces early dislocation. We have retrospectively reviewed elder patients who underwent total hip arthroplasty (THA) with cemented double mobility cup between March 2009 and January 2018. The inclusion criteria were patients (>75 years) who were operated on for primary THA (osteoarthritis or necrosis) with a cemented dual mobility cup and a high-risk instability (at least two patient-dependent risk factors for instability). The exclusion criteria were revision surgeries or hip fracture. In all the cases, the same surgical approach was performed with a Watson Jones modified approach in supine position. We have collected demographic data, instability risk factors. Patients were classified using the Devane’s score, Merle d’Aubigné score and the patient’s likelihood of falling with the Morse Fall Scale. Surgical and follow-up complications were collected from their medical history. Sixty-eight arthroplasties (68 patients) were included in the study. The median age was 81.7 years (SD 6.4), and the American Society of Anesthesiologists (ASA) score showed a distribution: II 27.94%, III 63.24% and IV 8.82%. Devane’s score was less than five in all of the cases. At least two patient-dependent risk factors for instability (87% had three or more) were present in each case. The median follow-up time was 49.04 months (SD 22.6). Complications observed were two cases of infection and one case of aseptic loosening at 15 months which required revision surgery. We did not observe any prosthetic dislocation. The cemented dual mobility cup is an excellent surgical option on primary total hip arthroplasties for elder patients with high-risk instability.

## 1. Introduction

Dislocation in hip arthroplasty has an incidence that varies from 2%–5% in primary surgery to 10%–25% in revision cases [1]. Furthermore, it is the most common cause of revision in primary total hip arthroplasty (THA) with 22.5% of them in US Medicare population [2]. Usually, the dislocation occurs within the first 3 years since the primary surgical procedure [3].

Instable THA implies an important increment in the direct and indirect healthy cost, even more so in the elderly population. THA dislocation in the elderly population has consequences on mortality, with a sixfold higher mortality rate of 65% compared to a 10% mortality rate during the same period of those without dislocation.

Finally, THA instability is one of the most difficult complications to treat because of its multifactorial etiology [4]. These factors are related to the patient, the surgical technique and the prosthesis. Patient-dependent risk factors include advanced age, female gender, previous hip surgical procedures, neurological diseases (dementia and Alzheimer’s), neuromuscular diseases (Parkinson’s, poliomyelitis sequelae, myopathies) and the diagnosis for which surgery is indicated (avascular necrosis, hip fractures, hip dysplasia, inflammatory arthritis, tumors of the proximal femur, among others) [5,6,7]. One study of 2023 THAs found that the rate of dislocation was ten times higher in patients with a higher American Society of Anesthesiologists (ASA) score [8].

Different surgical options have been described to reduce the instability risk. The use of a liner augmentation wedge [9] or angle-bore component [10], jumbo head [11,12,13] and constrained liner have been used with limited success [4,10,12,13,14,15]. The dual mobility cup (DMC) is used with the primary aim to prevent arthroplasty dislocation [11,12,13,14,15].

The concept of DMCs was first described by Gilles Bousquet in France in 1976. It is a combination of a hemispherical metallic cup with a large insert of polyethylene containing a metal femoral head of 22 mm with a retained system. The femoral head–polyethylene insert complex behaves as a large femoral head, increasing the articular range, and the radius of the head–femoral neck and thus increasing the jump distance before the dislocation event [10,11,16,17].

Although DMCs have been used for several years in France, the FDA approved their use in the USA in 2009. Since then, the interest in these tripolar cups is increasing. Large numbers of original models offered by the major American and European companies are in use, with different options such as cemented or cementless cups.

Our aim is to assess the results in terms of function, complications (with a focus on dislocations) and survival rates of the cemented DMCs in primary arthroplasty in elderly patients (>75 years) with a high risk of instability.

## 2. Materials and Methods

We have retrospectively reviewed patients with high-risk instability who underwent primary total hip arthroplasty with cemented DMCs between March 2009 and January 2018 in our institution, with a minimum follow-up of 2 years. We have included patients with primary total hip arthroplasty due to osteoarthritis or necrosis and at least 2 instability risk factors. These factors have been described previously [4] and are summarized in Table 1. We have excluded revision surgeries, those secondary to hip fracture, patients who had less than two patient-dependent risk factors for instability and patients who had lumbar spine stabilization.

The age, sex, diagnosis, ASA score [18] (American Society of Anesthesiologists), previous surgical procedures, neurological disorders, surgical and follow-up complications were collected from their medical history following the risk factors for instability. The ASA score was determined by the anesthesia department of our center.

Each patient was assigned a Devane score according to their activity level. Furthermore, we collected a pre-operative and post-operative Merle d’Aubigné score and the Morse Fall Scale [19]. The Morse Fall Scale assess the patient’s likelihood of falling. It consists of six variables: history of falling, secondary diagnosis, ambulatory aids, intravenous therapy, gait and mental status). Scoring these variables, the scale assesses the risk level of falling (0–24, no Risk; 25–50, low risk; 51 or more, high risk) and recommended actions are then identified for each patient (e.g., no interventions needed, standard fall prevention interventions, and high-risk prevention interventions). These standardized scores show the functional outcomes of a fragile population. All of them were collected from their medical histories.

### 2.1. Surgical Procedure

All the surgeries were performed by the hip surgery team of our institution (J.L.E., A.V.A., P.D.R., M.S.J. and J.R.V.). The same surgical technique was performed with a Watson Jones modified approach in supine position under general anesthesia with antibiotic prophylaxis with cefazolin (2 g) (repeated doses of 1 g every 3 h from the start). In all cases, a cemented stem was used (Müller stem, Zimmer Biomet^®^ Warsaw, IN, USA). The distribution of cemented double mobility cups were 51 Novae (SERF^®^); 11 ADES (ZimmerBiomet^®^, Warsaw, IN, USA) and six Avantage (ZimmerBiomet^®^, Warsaw, IN, USA).

Follow-up visits were scheduled preoperatively and occurred postoperatively at 3, 6, and 12 months and yearly thereafter. All patients gave informed consent before study entrance.

### 2.2. Statistical Analysis

Statistical analysis was performed with Statistical Package for the Social Sciences (SPSS) software (version 22.0 for Windows; IBM, Armonk, NY, USA). The preoperative scores were compared with the last follow- up scores with the use of a paired Student’s *t*-test for parametric data. The end point criterion was cup revision for any reason.

## 3. Results

The initial sample of patients who underwent THA-DMC was 179. We excluded 66 patients operated by hip fracture and 45 operated of a revision surgery. After the inclusion and exclusion criteria, 69 hips were selected between March 2009 and January 2018 (Figure 1). We excluded one patient that did not assist to controls. Finally, 68 THAs were included, 53 women and 15 men. Thirty-nine were performed on the right side and 29 on the left side. The median age of the patients was 81.7 years (SD 6.4), and the ASA score showed a distribution: grade II 27.94%, grade III 63.24% and grade IV 8.82%. The median follow-up time was 49.04 months (SD 22.6) with a minimum follow-up of 2 years.

Seven patients (10.3%) are grade IV of Devane, 17 (25%) are grade III, 39 (57.2%) are grade II, and 5 (7.5%) are grade I. All cases have at least two prosthetic patient-dependent risk factors for dislocation, and in 59 (87%) cases, three or more risk factors were identified. The Merle D’Aubigne score mean improved from 10.31 (preoperative) to 15.61 (postoperative) (*t* Student *p* = 0.03) According to the Morse Fall Scale, seven patients (10.3%) are at low risk of falling, 36 (52.9%) are at intermediate risk of falling and 25 patients (36.8%) are at high risk of falling. The demographic data are summarized in Table 2.

No prosthetic dislocations were assessed. We found two cases of superficial wound infection, treated with debridement, antibiotic and retention of implant (DAIR) and one aseptic loosening at 15 months, which required revision cup surgery (3/68). No post-operative blood transfusions were needed. The mean post-operative hemoglobin was 11.5 mg/dL (SD 2.8) at 24 h.

## 4. Discussion

Dual mobility cups in primary THAs reduce dislocations. Nevertheless, doubts about their long-term results, function preservation and cost-effectiveness are still present [20,21,22,23]. In the present article, we report our experience in the prevention of dislocations, using a cemented dual mobility cup in primary THAs in a selected cohort of elder and frail patients with a high preoperative dislocation and fall risk.

Bouchet et al., in a case-control study on primary THAs in 75 year-old patients (105 patients with dual mobility cup against 108 patients conventional 28-mm metal-polyethylene cup) reported five dislocations in the conventional group and no dislocation cases in the dual mobility group using the posterior approach [9]. Our results in a selected cohort has been shown as effective.

The constrained cup and DM cup share indications. However, there are several reports of high rates of early loosening with constrained cups [24,25,26,27]. Failed constrained cups data range from 7.1% at 12 months [28] to 14% at 36 months [29]. The constraint system transmits all of the stress to the cup fixation; the force is absorbed by the acetabulum because the constraint system is transmitted to the bone–implant interface, inducing a shear stress which leads to loosening [30,31,32,33]. The mechanical opposition to the dislocation creates a risk of the bone implant tearing. This mechanism is not present in DMCs.

A particular complication of dual mobility cups is the intra-prosthetic dislocation [34], consisting of the dislocation of polyethylene (due to the loss of the retentive system) while the femoral head articulates with the metal dome, which causes severe metallosis in a short period of time [32]. Philippot et al., in 438 primary THAs with dual mobility cups, after 17 years reported 23 (5.2%) intra-prosthetic dislocations (2% within 10 years of follow up) [17]. It is probably that with the use of the new polyethylene these complications will be more infrequent, but further data are needed [33,34,35,36]. No patient in our series presents this complication, but we have to consider that cases of intra-prosthetic dislocation have been described from 8 to 10 years after surgery, since wear on the retentive mechanism of the polyethylene insert should occur before dislocation. Recently, with 3.6 years of follow up, Jones et al. described only one intra-prosthetic dislocation in 151 patients [37].

Analyzing the long-term results of the non-cemented series, some previously published doubts in survival rates are present, reporting that their use in young patients is still controversial [14,15,38,39,40]. In that sense, Leiber-Wackenheim et al. [41] reported 19% of periacetabular radiolucencies at follow-up after 8 years when an uncemented dual mobility cup was implanted in revision surgery. Philippot et al., in a recent revision of 130 first generation of dual mobility cups in patients under fifty years old, reported a survival of 77% in 20 years (44 hips were revised, including 21 cup aseptic loosenings isolated, and 15 intra-prosthetic dislocations) [14]. We have found one case of aseptic cup loosening that occurred 15 months after primary THA with a cemented double mobility cup. We interpret that premature loosening as an initial error in the cementation technique and not because a secondary osteolysis was induced by debris. Because of that, we understand that DM cemented cups are a better option for the elder population.

We have found a relatively high incidence of complications but, most of them are considered independent of the cup implanted. There were two cases of deep infections and one case of aseptic loosening at 15 months which required revision surgery. We should note that elderly patients have more comorbidity factors associated, as we have shown in THE ASA score, often with poor nutritional status which makes them a group of patients at high risk for developing an infection.

In our study group, all the patients had a score of Devane between 2 to 4, which means an activity spectrum from semi-sedentary to homecare activity. Some controversies have been reported about the use of cemented metal-backed cups because of concerns about the decreased polyethylene thickness and the potential for increased wear and higher cost against the polyethylene cemented cups [42,43]. In younger patients it should be a problem, however, in an elderly population, with the presence of osteoporotic bone, relatively low demand, and more than 5 years of life expectancy, we prefer the use of cemented DM-THA cups. In the same manner, although all polyethylene cups are cheaper, in the selected high-risk instability patients, DMC have been shown to have cost-effectiveness when compared to conventional bearings [44]. It has been estimated that using DM-THA would be expected to save EUR 28.3 million per 100,000 THAs [45].

The improvement in pain and function was of 5 points in the Merle D’Aubigné postoperative score. Furthermore, the Morse Fall Scale showed that 89.7% of the patients had an intermediate or high risk of falling, showing the need to reduce the rate of dislocations such as those made by DM cups. In this range of age, similar function data have been reported between conventional THAs and DMCs [3,46].

Our study is not exempt of limitations. We know that a major limitation of our work is the short initial follow-up, so we cannot asses the incidence of aseptic loosening in the medium and long term. Despite the fact that our follow up is short, the aim of the study was specifically to review the effect in the prevention of dislocations in elder high-risk patients which are most frequent during the first year [31,37]. The Scottish National Arthroplasty Project on 61,274 THAs reported an incidence of 0.9% in the first two years, with no increase in the rate of dislocation after two years. Our follow up do not differ from other series published previously.

## 5. Conclusions

Cemented dual mobility cups in primary THAs represent an excellent surgical alternative to control dislocation complications in elderly (>75 years) and frail patients with a high risk of instability.

## Figures and Tables

**Figure 1 geriatrics-06-00023-f001:**
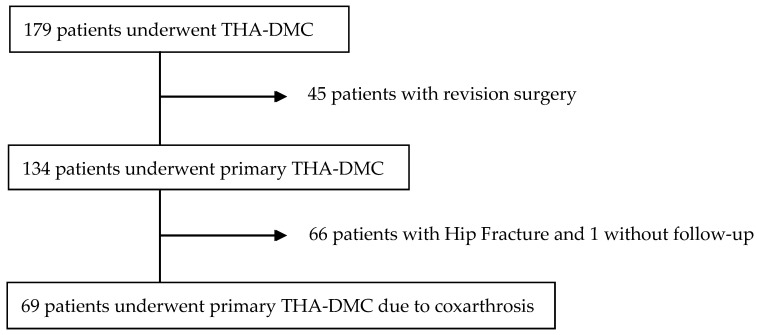
Flow diagram of patients selection. THA: total hip arthroplasty; DMC: dual mobility cup.

**Table 1 geriatrics-06-00023-t001:** Instability risk factors. ASA: American Society of Anesthesiologists.

Age ≥ 70 Years (Women) or ≥75 Years (Men)
Previous surgery of the hip; acetabular osteotomy, proximal femoral osteotomy, failed hip osteosynthesis.
Base diagnosis; hip dysplasia, femoral neck fracture, avascular necrosis of femoral head, inflammatory arthritis, proximal femoral tumors.
Neuromuscular disease; epilepsy, cerebral palsy, polio sequelae, Parkinson’s and myopathies
Cognitive dysfunction; dementia, Alzheimer’s, alcoholism
ASA score ≥ 3
Revisions of primary arthroplasty

**Table 2 geriatrics-06-00023-t002:** Demographic data and baseline characteristics of the study population.

Participants	*n* = 68 Patients
Age (years (SD))	81.7 (6.4)
Sex (Female/Male)	53/15
Right/Left	39/29
ASA (Grade (%))	I 0%; II 27.94%; III 63.24%; IV 8.82%
Dislocation Risk factors	Two factors: 9 Three factors: 59
Morse Fall Scale	Low risk: 7 Intermediate risk: 36 High risk: 25
Complications (3/68)	Superficial wound infection: 2 Aseptic loosening; 1 Dislocations: 0
Follow-up (mean (SD))	49.04 months (22.6)

## Data Availability

The data presented in this study are available on request from the corresponding author.

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
