# Peer review of "Cemented Dual Mobility Cup for Primary Total Hip Arthroplasty in Elder Patients with High-Risk Instability"

_geriatrics, 2021, doi:10.3390/geriatrics6010023_

Round 1
Reviewer 1 Report
Thank you very much for the opportunity to review this very interesting study.
Summary: The authors retrospectively analyzed 68 elderly patients (age >75yo) treated with a cemented dual mobility cup for primary THA deemed high-risk by two patient risk factors for instability. Median follow-up time was 49.04 months, and they found three negative outcomes: two infections and one aseptic loosening; all requiring reoperation. Authors concluded that this treatment was an excellent option for elderly patients with high-risk instability.
Pros: High-risk elderly patients are at an increased risk for dislocation after primary THA, and this study focuses on this important topic by analyzing a surgical modality to decrease these outcomes. Very thorough introduction and discussion of the literature.
Cons: My biggest concern is the lack of a control group. The authors state that their 3 cases of complications is relatively high, but have no comparison for these patients that represent a higher risk of complications.
Major comments:
Language editing for grammar and spelling is required for clarity and flow of author ideas.
No control group, thus it is hard to determine the significance of the DMC and the rate of reoperations.
Authors state the median follow-up time was 49.04 months (SD 22.6), but do not state the minimum follow-up. Generally, TJA studies would like a minimum of 1-2 follow-up for cases.
Minor comments:
A table representing the statistical tests you ran would help summarize (eg the Merl D' Aubigne score
Putting a percentage on the reoperation rate would help discuss your study in the context of others (eg 3/68).
Extensive editing of English language and style required.
Author Response
Dear Reviewers.
We have changed the submission according to your comments. Also, we have reviewed the English and reduce the bibliography.
King regards
Reviewer 2 Report
The study actually confirms the value of DMC.
The disadvantage of the study is the short observation period.
Comments:
Introduction
1st paragraph can be removed - an obvious sentence.
2. Please shorten the information about DMC.
In the inclusion criteria, it is worth noting whether the patients had lumbar spine stabilization. It also increases the risk of THA instability
Table 2. Sex (Male / Female) is the opposite of what is stated in the text
It was not stated who assessed the ASA
Too many references - it would be nice to reduce to 30.
The first 11 references are over 10 years old.
English correction
desbridamient -> debridement
Suvirval -> survival
Author Response

(The authors gave the same response as above.)

Reviewer 3 Report
Dear Authors,
Below is a point by point review of the manuscript.
Title: good
English: changes are required throughout, for example in the abstract, line 14: ‘Several studies have proved that dual mobility cups (DM) reduces.’
Abstract: some corrections are expected in line with manuscript body changes.
Key words: good
Introduction: good
Material and method: inclusion criteria are difficult to understand; you include patients with at least 2 criteria from the those in the table? When you decide to put a cemented dual mobility cup in the first place one would expect the patient to be old and have serious risk of instability.
Lines 85-86: reference [8] is unclear. When searched on google it is not an 2009 instructional course but a 2008 JBJS title. Also, instability risk factors are not universally agreed upon and are the core evidence of your selection. Therefore a much more detailed presentation and evidence to support them is needed.
Lines 92-99: please also briefly detail the Devane score and the Merle d´Aubigné and their utility; how did you assign the scores?
Lines 101-102: are probably incomplete. Blood management is presented confuse and not important.
Lines 120-121: should not be in the statistical analysis paragraph.
Ethics: no ethics are discussed or presented; no mention of raw data availability
Statistics: you only compared the pre op to postop function scores and found, as expected that hip function improved after joint replacement.
Results: no comparative statistic results are presented; we do not know if these patients would have dislocated with a primary cemented implant. Please give more data on time of follow-up; you have a mean of 49 months with a very big SD of 22. Could you find a control cohort of age gender matched patients with conventional implants? Even though they may not have as many instability risk factors. You could at least do a regression analysis and look for potential associations between the instability factors and the 3 scores.
Discussion: good
Conclusion: All we find in your study is that over 10 years you selected very well the patients for a cemented dual mobility hip replacement which led to good outcomes.
References: some require attention
Tables and pictures: Tab. 1 should be revised; the title in bold is just one factor of instability; maybe just list them and not use a table. Table 2. Please add details on instability risk factors.
Overall: interesting and potentially useful. However major revision is required to bring out the full potential.
Looking forward to your resubmission.
Sincerely,
Author Response

(The authors gave the same response as above.)

Round 2
Reviewer 1 Report
My comments were appropriately addressed.
